# Low-Cost and High-Performance Solution for Positioning and Monitoring of Large Structures

**DOI:** 10.3390/s22051788

**Published:** 2022-02-24

**Authors:** Giorgio de Alteriis, Claudia Conte, Enzo Caputo, Paolo Chiariotti, Domenico Accardo, Alfredo Cigada, Rosario Schiano Lo Moriello

**Affiliations:** 1Department of Industrial Engineering, University of Naples Federico II, Piazzale Tecchio 80, 80125 Naples, Italy; claudia.conte2@unina.it (C.C.); enzo.caputo@unina.it (E.C.); domenico.accardo@unina.it (D.A.); rschiano@unina.it (R.S.L.M.); 2Department of Management, Information and Production Engineering, University of Bergamo, 24044 Dalmine, Italy; 3Department of Mechanical Engineering, Politecnico di Milano, 20156 Milano, Italy; paolo.chiariotti@polimi.it (P.C.); alfredo.cigada@polimi.it (A.C.)

**Keywords:** position and attitude estimation, MEMS sensors, inertial measurement unit, GPS-RTK correction, Kalman filtering

## Abstract

Systems for accurate attitude and position monitoring of large structures, such as bridges, tunnels, and offshore platforms are changing in recent years thanks to the exploitation of sensors based on Micro-ElectroMechanical Systems (MEMS) as an Inertial Measurement Unit (IMU). Currently adopted solutions are, in fact, mainly based on fiber optic sensors (characterized by high performance in attitude estimation to the detriment of relevant costs large volumes and heavy weights) and integrated with a Global Position System (GPS) capable of providing low-frequency or single-update information about the position. To provide a cost-effective alternative and overcome the limitations in terms of dimensions and position update frequency, a suitable solution and a corresponding prototype, exhibiting performance very close to those of the traditional solutions, are presented and described hereinafter. The solution leverages a real-time Kalman filter that, along with the proper features of the MEMS inertial sensor and Real-Time Kinematic (RTK) GPS, allows achieving performance in terms of attitude and position estimates suitable for this kind of application. The results obtained in a number of tests underline the promising reliability and effectiveness of the solution in estimating the attitude and position of large structures. In particular, several tests carried out in the laboratory highlighted high system stability; standard deviations of attitude estimates as low as 0.04° were, in fact, experienced in tests conducted in static conditions. Moreover, the prototype performance was also compared with a fiber optic sensor in tests emulating actual operating conditions; differences in the order of a few hundredths of a degree were found in the attitude measurements.

## 1. Introduction

In recent years, a growing interest in solutions for monitoring the attitude and position of large structures (especially bridges, tunnels, and offshore platforms) has pervaded both scientific and industrial research [1,2,3]. As an example, the field monitoring of offshore structures allows raw data to be obtained directly and in real-time, thus enabling the timely detection of structural failures, safety assessments, predictions of performance changes, and remaining structure lifetime [4]. In addition, field monitoring can verify design parameters and provide a database for post-project analysis [5]. Due to the complex environmental stresses and failure mechanisms of the riser and mooring systems, field monitoring has become an effective method to obtain real-time tracking and feedback information based on a specialized monitoring system to reduce the risk of failure. Besides long-term monitoring, field monitoring has been proven to be a necessary operational support also during platform installation over the sea [6].

Currently exploited techniques are mainly based on Global Navigation Satellite Systems (GNSSs) and Inertial Measurement Units (IMUs) [7,8], which use a double integration process of raw accelerometer and gyroscope readings to estimate position and attitude. Different integrations and operational approaches have already been presented in the literature for both GNSS and IMU [9]. As an example, the offshore structure position was measured by means of a differential GNSS signal in [10]. In particular, two GNSS modules were installed on the structure and one at the base station, in such a way that the three-dimensional coordinates were exploited to obtain the trajectory and relative geographic north position of the structure with high accuracy. This method is based on the Global Positioning System (GPS) baseline and the angle it forms with the geographic north as described in [11]. The solution adopted in [12] exploits, instead, a particular GNSS service, the so-called real-time service (RTS), which uses Short Message Communication (SMC) based on the BeiDou navigation satellite system to retrieve the measures of longitudinal and transverse track correction, radial direction, and clock needed to evaluate only the position of the offshore structure in static and dynamic conditions. Examples of GNSS/IMU integration are described in [13,14], where accelerometers and gyroscopes were used to monitor the inclination and orientation of the structures with advantages in terms of noise immunity to external influences. A Real-Time Kinematic (RTK) GNSS has been used as a correction stage for the typical errors that affect inertial sensors [15,16,17,18]; a differential GNSS approach was exploited to achieve higher accuracy. In particular, the two mobile stations consisted of a laser and a fiber-optic inertial sensor integrated into their own GPS module and shared the base reference station to take advantage of GNSS and inertial sensors [19].

Most of the considered solutions mainly leveraged the integration of fiber optic inertial sensors and GPS RTK systems. The performance of these systems is ensured by using the best inertial sensor technologies, i.e., Fiber Optic Gyroscope (FOG), which allows achieving high performance in terms of accurate attitude estimates [20,21]. Thanks to the development of novel sensors based on microelectronic fabrication techniques, it is now possible to obtain high-performance systems that exploit the advantages of MEMS sensors, such as low cost, small size, low power consumption, and high integration capability. On the contrary, inertial sensors manufactured in MEMS technology present characteristic errors, such as bias instability and drift, which need better compensation with respect to their conventional counterparts [22].

Based on the discussed solutions and to overcome the considered limitations of MEMS sensors, the authors propose a high accuracy real-time solution to monitor the attitude and position of structures based on the integration in a fifteen-state Kalman filter of an IMU involving a MEMS tri-axial accelerometer and tri-axial gyroscope and an RTK GNSS module. With the aim of evaluating the performance of the proposed solution, a prototype monitoring system was realized, consisting of two modules. The first module, installed on the structure to be monitored (e.g., bridges, tunnels, offshore platforms, and similar), consists of a microcontroller responsible for acquiring and processing the data from the IMU and GPS RTK module through an error-state Kalman filter and returning the attitude and position estimates. The second module mainly provides the RTK correction and acts as a wireless interface between the first module and the eventual final user.

This paper is organized as follows: the proposed solution along with the prototype developed for its assessment are described in Section 2; the results obtained in the preliminary characterization carried out in the laboratory are discussed in Section 3, while Section 4 focuses on the results of tests involving the comparison of the proposed prototype with a typical solution exploited for the considered application. The concluding remarks are finally given in Section 5.

## 2. Proposed Solution

The proposed solution aims at developing a low-cost embedded positioning and monitoring system based on the use of GPS modules in the RTK configuration and MEMS sensors, integrated by means of an Error-State Kalman Filter (ESKF), to estimate the attitude and position of large structures, such as bridges, tunnels or jackets for offshore platforms, in real time. The block diagram of the proposed solution is described in Figure 1, where *I = x, y,* and *z* represent the three sensing axes.

In order to gain the desired estimates, the first step consists of measuring the raw values of acceleration, *a_i_*, and angular rate, *ω_i_*, by means of inertial sensors, whose performance in terms of bias and drift have to be accurately chosen. As stated above, attention has been focused on so-called tactical-grade sensors, i.e., gyros and accelerometers characterized by bias instability lower than 0.1°/h and 60 μg, respectively [20]. Thanks to the traditional mechanization equations [15], raw acceleration and angular rate values are used to provide a first estimate of the desired quantities. For the sake of clarity, the mechanization equations needed to gain the navigation solution are presented in the following in their continuous form; the actual version implemented in the solution is obtained by means of the time discretization. According to the traditional navigation solution, the subscripts and superscripts refer to the different involved reference frame; in particular, *b* stands for the body frame, *I* indicates the inertial frame, and *n* is the local navigation frame (also referred to as NED: North, East, and Down).

The process can be organized into four steps:1.The attitude update is first computed through the time derivative of coordinate transformation matrix $C_{b}^{n}$ (body to navigation reference frame) at time t according to
(1)$${\dot{C}}_{b}^{n}=C_{b}^{n}(\Omega_{ib}^{b}-b_{g})-\left(\Omega_{ie}^{n}+\Omega_{en}^{n}\right)C_{b}^{n}$$
where $\Omega_{ib}^{b}$ is the skew-symmetric matrix of the IMU angular rate measurements ($\omega_{ib}^{b}=(\omega_{ib_{x}}^{b},\omega_{ib_{y}}^{b},\omega_{ib_{z}}^{b}$) corrected from the estimated bias value, $b_{g}$
(2)$$\Omega_{ib}^{b}=\left[\begin{array}{ccc} 0 & -\omega_{ib_{z}}^{b} & \omega_{ib_{y}}^{b}\\ \omega_{ib_{z}}^{b} & 0 & -\omega_{ib_{x}}^{b}\\ -\omega_{ib_{y}}^{b} & \omega_{ib_{x}}^{b} & 0 \end{array}\right]$$

The matrix $\Omega_{ie}^{n}$ represents the skew-symmetric matrix of the Earth’s rotation (depending on the latitude and the Earth’s angular rate), while $\Omega_{en}^{n}$ is the skew-symmetric matrix of the transport rate (depending on the system velocity, latitude, altitude, and the Earth surface’s radius of curvature).

2.Specific force frame transformation that allows computing the measured specific force (acceleration) of the body with respect to inertial space resolved in the current local navigation frame(3)$$f_{ib}^{n}=C_{b}^{n}(f_{ib}^{b}-b_{a})$$where $f_{ib}^{b}=\left(f_{ib_{x}}^{b},f_{ib_{y}}^{b},f_{ib_{z}}^{b}\right)$ is the specific force sensed by the accelerometers; also, in this case, measured accelerations must be compensated from possible biases, $b_{a}$.
3.The velocity vector $v_{eb}^{n}=(v_{eb_{N}}^{n},v_{eb_{E}}^{n}$, $v_{eb_{D}}^{n})$ is then updated by computing the time derivative of velocity at time t according to the following expression,
(4)$$\dot{v_{eb}^{n}}=f_{ib}^{n}+g_{in}^{b}-\left[\Omega_{en}^{n}+2\Omega_{ie}^{n}\right]v_{eb}^{n}$$where $g_{in}^{b}$ is the acceleration due to gravity evaluated according to [15].
4.The position in the local navigation frame is finally updated by computing the time derivative of latitude ($L_{b}$, longitude ($\lambda_{b}$), and altitude ($h_{b}$) at time t according to the following relationships
(5)$$\dot{L_{b}}=\frac{v_{eb_{N}}^{n}}{R_{N}\left(L_{b}\right)+h_{b}}$$
(6)$$\dot{\lambda_{b}}=\frac{v_{eb_{E}}^{n}secL_{b}}{R_{E}\left(L_{b}\right)+h_{b}}$$
(7)$$\dot{h_{b}}=-v_{eb_{D}}^{n}$$where $R_{N}$ and $R_{E}$ are respectively the meridian and transverse radius of the curvature of the Earth’s ellipsoidal surface.

However, the residual errors of bias and drift make the estimates diverge from the nominal values for increasing time intervals; this way, the bias values of gyros and accelerometers must be continuously updated. To this aim, the solution leverages the exploitation of a loosely coupled ESKF to integrate measurements of position, *P_GPS_*, and velocity, *v_i_*, provided by the GPS module [15,23,24,25,26,27,28]. According to the Kalman filtering approach, a priori estimates of the errors are periodically improved in the correction stage thanks to the position and velocity measurements obtained from the GPS module. The filter outputs are used as correction values in the mechanization equations, and Δ***V***, Δ***P***, and Δ*bias* represent the correction values of velocity, position, and biases (accelerometer and gyroscope), respectively, among the three axes.

As for the implementation of the integrated IMU/GPS RTK navigation system, the state vector to be estimated strictly depends on the IMU state vector and then includes 15 state variables organized into five 3-dimensional vectors representing attitude (ΔΩ), velocity (ΔV**,** in the NED reference frame), and position (ΔP) errors, as well as accelerometer and gyroscope biases, ($b_{a}$, $b_{g}$ respectively). For the sake of clarity, the state vector $\hat{x}$ of the ESKF is reported in the following:(8)$$\hat{x}=\{\begin{array}{c} \begin{array}{c} \Delta \Omega =\left(\Delta \psi ,\;\Delta \theta ,\;\Delta \varphi \right)\\ \Delta V=\left(\Delta V_{N},\Delta V_{E},\delta V_{D}\right)\\ \Delta P=\left(\Delta latitutde,\;\Delta longitude,\Delta altitude\right) \end{array}\\ b_{a}=\left(b_{ax},b_{ay},b_{az}\right)\\ b_{g}=\left(b_{gx},b_{by},b_{gz}\right) \end{array}$$

Since a loosely coupled integration was chosen, no GPS RTK state variables are considered and estimated.

As stated above, the prediction and correction algorithm adopted is based on the Kalman Filter as described in [15]. In particular, it is based on two equations groups, referred to as Time and Measurement Update, respectively, and involved in the propagation (evaluation of a new state vector estimation) and correction (estimation improvement thanks to the availability of external measurements) stages. In the following equations, the subscript j denotes the generic iteration index, whereas the symbols − and + indicate the propagation stage and the correction stage, respectively.
1.Each time new values of acceleration and angular rate are available, the prediction or “Time Update” equations are exploited for the propagation of both the state vector and state error covariance matrix
(9)$${\hat{x}}_{j}^{-}=F\;{\hat{x}}_{j-1}^{+}$$
(10)$$P_{j}^{-}=F\;P_{j-1}^{+}F^{\prime }+Q\;$$
where F is the system model matrix whose elements are fully described in [15] and are resolved in the local navigation frame. The system model matrix describes how the system evolves over time and allows gaining the so-called a priori estimate of the state vector. Instead, P is the error covariance matrix that takes into account the system noise by means of the system noise covariance matrix, Q, that defines the main noise sources of the inertial sensors; in particular, the entries of the matrix Q can be set starting from the bias instability and random walk available in the inertial sensors datasheet.
2.Each time a new vector ***z****_j_* of position and velocity is available from the GNSS, the correction or “Measurement update” equations are exploited to provide an improved estimate (usually referred to as a posteriori) of the state vector. For this, the Kalman gain (K), i.e., a parameter weighting the reliability of the information introduced by the external measurement, is first evaluated according to
(11)$$K=P_{j}^{-}H_{j}^{\prime }{\left(H_{j}P_{j}^{-}H_{j}^{’}+R_{j}\right)}^{-1}\;$$
where H is the measurement matrix that describes how the measurements are deterministically related with states and in a loosely coupled integration approach that can be approximated as described in [15]
(12)$$H=\left[\begin{array}{cc} 0_{3} & 0_{3}\\ 0_{3} & -I_{3} \end{array}\begin{array}{ccc} -I_{3} & 0_{3} & 0_{3}\\ 0_{3} & 0_{3} & 0_{3} \end{array}\right]$$
where $0_{3}$ indicates a 3 × 3 matrix whose entries are all equal to 0, while $I_{3}$ stands for the 3 × 3 identity matrix; in the determination of the ***H*** entries, no GPS states are estimated and the lever arm (i.e., distance from IMU to GPS antenna) contribution is neglected due to the weak coupling of the attitude errors and gyroscope bias, excepts where the level arm is very large.

R is the noise covariance matrix of GPS measurements, i.e., a diagonal matrix whose elements are the variance of the measurement noise component evaluated as an independent source of white noise, i.e., providing a statistical characterization of the measurement noise properties.

The a posteriori estimate of the state vector along with the state error covariance matrix is then achieved thanks to the following expressions
(13)$${\hat{x}}_{j}^{+}={\hat{x}}_{j}^{-}+K_{j}\left(z_{j}-\hat{z_{j}}\right)\;$$
(14)$$P=\left(I-K_{j}H_{j}\right)P_{j}^{-}$$
where $\hat{z_{j}}$ is the position and velocity estimated in Equations (4)–(6).

The a posteriori estimates of the state vector provide the information needed to correct the position, velocity, and attitude of the mechanization equations, as well as the bias values for acceleration and angular rate compensation. Before starting a new cycle of the Kalman filter, all components of the state vector are zeroed but the biases.

The proposed solution is based on the use of GPS RTK modules that simultaneously ensure centimeter-level accuracy and a higher data rate (20 Hz) than the traditional module (typically equal to 1 Hz). To this aim, an external module has to be considered, capable of communicating correction values for the measurements of position, Δ*P_RTK_*, and velocity, Δ*V_RTK_*, exploited in the Kalman filter. Thanks to the considered approach, the solution is finally capable of providing the desired values of position, *P*, and attitude, i.e., the heading, *ψ*, pitch, *θ*, and roll, *φ*, angles of the structure, thus making possible its monitoring during both the installation and operating stages.

Similar considerations hold for the preliminary bias-estimation stage, carried out by means of a Zero-Velocity Update (ZUPT) filter [15,29,30,31,32,33]; the ZUPT is also based on Kalman filtering and exploits a known standing condition to evaluate the accelerometer, *ba_i_*, and gyro biases, *bg_i_*, responsible for the experienced differences between the standing values of attitude and velocity and the solution outputs.

A prototype implementation of the proposed solution has been developed in order to properly assess its performance. The prototype mainly consists of two modules; as can be appreciated in Figure 2, the modules share most of the hardware architecture and components. In particular, the first module, referred to as Rover station and equipped with both the GPS RTK module and IMU, is installed on the structure to be monitored. The Rover is mandated to implement the procedure for position and attitude monitoring and send the measured values to the second module, referred to as the Base station. The Base Station is mandated to receive the measured data, provide them to the interested user, and send the RTK correction associated with its own GPS module to the Rover. 

As for the implementation of each module, the Rover station embeds a MEMS sensor, a GPS RTK module, a microcontroller, a power supply, and a Bluetooth communication module. In particular,
The selected IMU is a tactical-grade MEMS sensor ADIS16495 from Analog Device^TM^ that includes, as the sensing unit, both a triaxial digital gyroscope and a triaxial digital accelerometer The gyroscope is characterized by a bias instability equal to 0.8°/h and an angular random walk equal to 0.09°/√h; as for the accelerometer, the datasheet [34] describes a bias instability equal to 3.2 μg and velocity random walk equal to 0.008 m/s/√h. The gyro bias instability sets the selected sensor in the lower bound of the tactical-grade category and represents a proper trade-off between performance and costs;The GPS module adopted was ZED-F9P by Ublox^TM^ [35] included in the SparkFun^TM^ GPS-RTK board. The GPS module is equipped with two UART communication serial ports; the first one is demanded to retrieve the RTK correction, and the second one is configured to send position outputs to the microcontroller.Communications between the Rover and Base stations are implemented with a certified Bluetooth module by Roving Network^TM^ included in the SparkFun^TM^ WRL-12,580 that allows obtaining a stable connection within a range of 100 m and baud rate of up to 921,600 bps for data transmission [36].The core of the system was a Nucleo-F446 board from STMicroeletronics^TM^ with a Cortex-M4-based microcontroller that operates at 180 MHz [37]. The microcontroller of the Rover station is programmed in such a way as to initialize the SPI and UART peripherals and then establish a Bluetooth communication with the Base station. Once the connection is ready, the microcontroller waits for the acquisition start command. In addition, the microcontroller is programmed to check if the warm-up time has elapsed and then start the attitude and position monitoring procedure described above and in Figure 1. Finally, the results are sent to the Base station until a stop command is received. For the sake of clarity, the program executed by the microcontroller in the Rover station is summarized in the flow chart shown in Figure 3.The power source is granted by a lithium battery of 22,000 mAh.

The core system communicates with the ADIS16495 through the SPI protocol and with GPS and Bluetooth modules by means of two UART connections Moreover, in order to realize the GPS RTK correction, the GPS module is connected to an additional Bluetooth module; the developed prototype is presented in Figure 4.

The Base station is composed of a GPS RTK module, a microcontroller, and a communication module. The hardware components are the same for both modules; the main hardware differences consist of (i) the sensing unit that is assembled only in the Rover-station, and (ii) the power source of the Base-station that is obtained from the Personal Computer.

## 3. Prototype Assessment in Laboratory Tests

The realized system was preliminarily tested under two different conditions involving either a continuous, controlled rotation or a comparison with a high-performance fiber-optic gyroscope. Before assessing the prototype performance under each test condition, a suitable warm-up time, equal to 30 min, was allowed to elapse for the achievement of the thermal regime.

### 3.1. Test Conducted in Stationary Conditions

With the aim of preliminarily assessing the temporal stability of the solution-estimated outputs, a first test was carried out in stationary conditions and in a climatic chamber at a constant temperature of 25 °C; the prototype was roughly aligned in the north direction, and raw measurements from the MEMS IMU were acquired during an observation interval equal to ten minutes. The prototype position and attitude were estimated by means of the Kalman filter-based data fusion algorithm, and the obtained results in terms of estimates of roll, pitch, and heading are shown in Figure 5.

For the sake of clarity, the mean value, standard deviation, and root-mean-square (RMS) of the attitude angles are reported in Table 1; the obtained results highlight satisfactory stability within the considered observation interval.

### 3.2. Test Conducted in Controlled Rotations

Further tests involved controlled rotations carried out by means of a robotic arm, namely a KUKA^TM^ KR 120 R2700, characterized by a rated repeatability of 0.05 mm [38]; the corresponding experimental setup is shown in Figure 6. Two different tests were conducted; in the first, the prototype was rotated within a range from 0° to 360° with a step of 10° around the *z*-axis. After each step, the attitude of the prototype was estimated for one minute to assess the solution stability; in particular, the realized prototype was mounted on the robotic arm in such a way that its *z*-axis was aligned with the rotation axis of the arm. The obtained results in terms of the heading angle estimates are shown in Figure 7.

As for the stationary conditions, the prototype confirmed the performance assessed in the first laboratory tests; standard deviations of the acquired values as low as 0.03° were experienced when the prototype was held in a defined angular position.

Moreover, the rotation between successive angular positions was evaluated by the difference in the mean values of the corresponding heading estimates. The obtained results are shown in Figure 8 and numerically summarized in Table 2; an average measured rotation and associated standard deviation equal to 10.04° and 0.18°, respectively, assured performance compliant with the considered application.

Similar results were obtained in a successive experiment designed to assess the stability of the prototype when it was subjected to complete overturns. In particular, Figure 9 shows the evolutions of the estimated heading angles versus time; also, in this case, mean values of the differences between the nominal and measured angles under stationary conditions as low as 0.01° were encountered with a corresponding standard deviation equal to 0.15° (Table 3).

## 4. Prototype Assessment under Emulated Operating Conditions

The results of the tests carried out in the laboratory highlighted the suitable performance and stability of the developed system. In order to validate the proposed solution, the position and attitude estimates of both the realized system and a reference were compared under conditions as close as possible to the actual ones, i.e., jacket movement for offshore platform installation; to this end, both systems were mounted on an oscillating platform, as shown in Figure 10, emulating a typical jacket of an offshore platform. A fiber optic gyroscope typically adopted for this kind of application was selected as the reference. The adopted reference system was an Octans from iXBlue^TM^ [39] that presents a heading accuracy equal to 0.1°/cos (Latitude) and 0.01° for the pitch and roll angles, respectively. In particular, the prototype was tested under two different conditions: in the first one, the performance under stationary conditions was assessed, while, in the second one, the prototype and the reference captured a motion emulating a real operating condition composed of random tilt movements and arbitrary controlled rotations.

### 4.1. Comparison Tests: Stationary Conditions

In the first test, the attitude estimates of both systems were evaluated under standing conditions to observe the stability of the system and, in particular, the drift of the heading angle over time. 

For this purpose, the attitude estimates obtained from both systems within an interval of 6 min were observed. The comparison results are shown in Figure 11; for the sake of brevity, only the results regarding heading will be shown, and similar remarks apply to the pitch and roll angles. An initial phase, whose duration was about 2 min, was characterized by differences between the estimates provided by the proposed prototype and those granted by the reference equal to about 0.2°. Subsequently, the heading angle estimates of the proposed solution reached an average difference as low as 0.03°.

Thus, the prototype was stable after a short time interval (about 2 min) and achieved performance in terms of the mean, standard deviation, and root-mean-square-error (RMSE) of the differences between the estimates of the proposed solution and the reference sensor equal to 0.03°, 0.07°, and 0.08°, respectively. Similar results were obtained for the pitch and roll angles; the corresponding results are summarized in Table 4.

To better appreciate the prototype performance, the results of the comparison between the two systems expressed in terms of relative percentage differences for the heading angle are reported in Table 5. 

The results obtained in this test confirmed the performance obtained in the preliminary tests carried out in the laboratory during the standing conditions; in fact, the relative percentage error was about 0.013%. In the end, the values obtained from the comparison of the two systems showed that it is possible to obtain comparable attitude estimates; the greatest differences (in any case lower than 0.1%) were experienced in the initial short transient phase, associated with the Kalman filter convergence, and usually expired in 2 min.

### 4.2. Comparison Tests: Dynamic Conditions

Further tests were carried out to assess the prototype performance under dynamic conditions emulating real operations. To this aim, two different dynamic conditions were taken into account. The attitude angles of both prototype and reference systems were estimated and compared within an observation interval of about 10 min. In particular, after the initial phase of standing conditions mandated to assure the filter convergence (highlighted in gray in Figure 12), the platform was manually rotated in such a way as to alternate low-rate (highlighted with a green background in Figure 12) and high-rate (highlighted with an orange background in Figure 12) rotations.

The obtained results highlighted a remarkable concurrence between the estimates provided by the proposed solutions and those granted by the reference sensor. The comparison was carried out by evaluating the mean, standard deviation, and RMSE of the differences between the measured and reference angles; values equal to 0.05°, 0.38°, and 0.38°, respectively, confirmed the satisfying performance of the proposed solution. Similar results were also obtained for pitch and roll angles; the corresponding values for each attitude angle are summarized in Table 6.

As for the first test, the results of heading angle are also reported in terms of relative percentage differences in Table 7 to make the comparison easier. Differences as low as 0.6% were experienced throughout the test, confirming the promising performance of the prototype.

Finally, the position estimates were evaluated and compared with the latitude and longitude obtained with a topographical technique as a reference. The results obtained are shown in Figure 13 for latitude and longitude. The numerical results are reported in Table 8 in terms of the difference between the mean values obtained from the prototype and reference (Δ Position).

The results showed estimated position values lower than 5 × 10^−6^ decimal degrees, which correspond to 0.16 m for latitude and 0.54 m for longitude. The values obtained have shown the system’s capabilities not only for attitude estimation, but also for position estimation at a higher frequency, in this test at 125 Hz, that allow evaluating the offshore platforms’ dynamics during the positioning phase keeping the decimeter accuracy given by the GPS RTK system.

## 5. Conclusions

A new solution based on the integrated use of a tactical MEMS sensor and GPS RTK module for the accurate measurement and monitoring of the position and attitude of large structures was presented. The method required two communicating modules, mainly mandated to (i) carry out measurements of acceleration and angular rate and (ii) provide correction terms. More specifically, the module (Rover station) installed on the structure to be monitored, referred to as the rover station, must be equipped with the inertial sensors, both the accelerometer and gyro, and leverages an Error-State Kalman Filter to improve the position and attitude estimates gained thanks to traditional mechanization equations. To this aim, the Base station (i.e., the second considered module) transmits to the rover position and velocity measured by means of a GPS system with RTK correction at a suitable rate.

A prototype was implemented to preliminarily assess the method’s performance in tests carried out in the laboratory or emulating actual operating conditions. As for the Rover station, a MEMS inertial sensor, characterized by a fair trade-off between performance and costs, was exploited. The sensor was connected to a microcontroller mandated to manage data acquisition and processing according to the above-mentioned Error-State Kalman Filter. Communications between the Rover and Base station were realized via the Bluetooth protocol for both the RTK correction and system outputs, i.e., position and attitude estimates.

The prototype developed was preliminarily characterized in the laboratory to assess its time stability during ten-minute steady-state acquisition; values of deviation as low as 0.06° were experienced. Further tests involved ten-degree controlled rotations of the Rover station by means of a robotic arm; an average rotation of 10.04° was measured, with an associated standard deviation equal to 0.18°. The performance of the prototype was then compared with a FOG sensor, taken as the reference sensor, under emulated operating conditions; in particular, tests were carried out on a freely moving platform emulating a typical jacket for offshore platforms. The first static test confirmed the performance of the prototype developed; differences between the heading estimates provided by the prototype and those granted by the reference as low as 0.05° were experienced. Similar results were obtained for the other angles, with a corresponding relative difference expressed in percentage terms lower than 0.03% after Kalman filter convergence. Dynamic tests, alternating low-rate and high-rate rotations, were successively carried out; differences between the measured and reference heading always lower than 0.4% were encountered. Moreover, position estimates were evaluated with respect to the reference point, resulting in differences in the order of a few decimeters for latitude and longitude, thus confirming the reliability and efficacy of the proposed method and prototype.

## Figures and Tables

**Figure 1 sensors-22-01788-f001:**
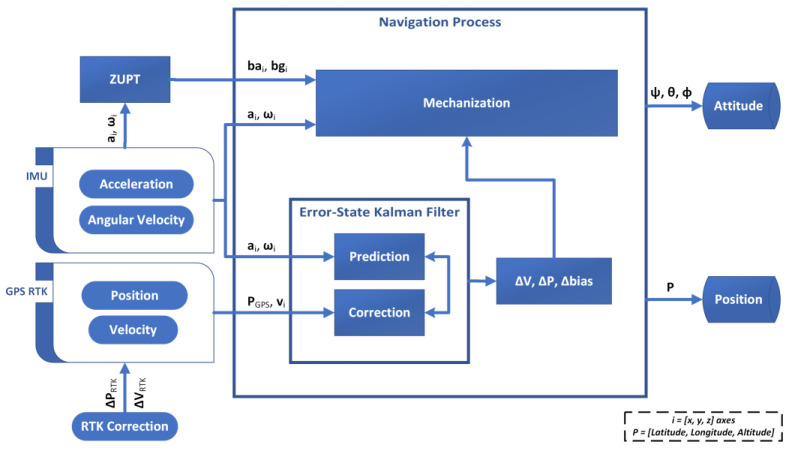
Block diagram of the proposed method based on GPS RTK/IMU integration for attitude and position monitoring.

**Figure 2 sensors-22-01788-f002:**
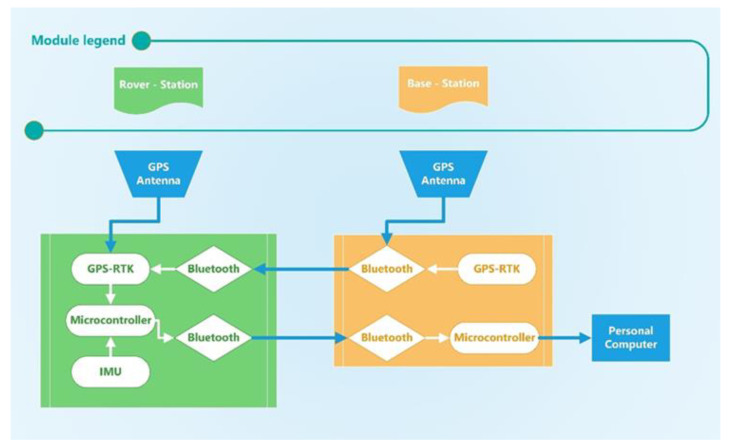
Block diagram of the proposed hardware architecture.

**Figure 3 sensors-22-01788-f003:**
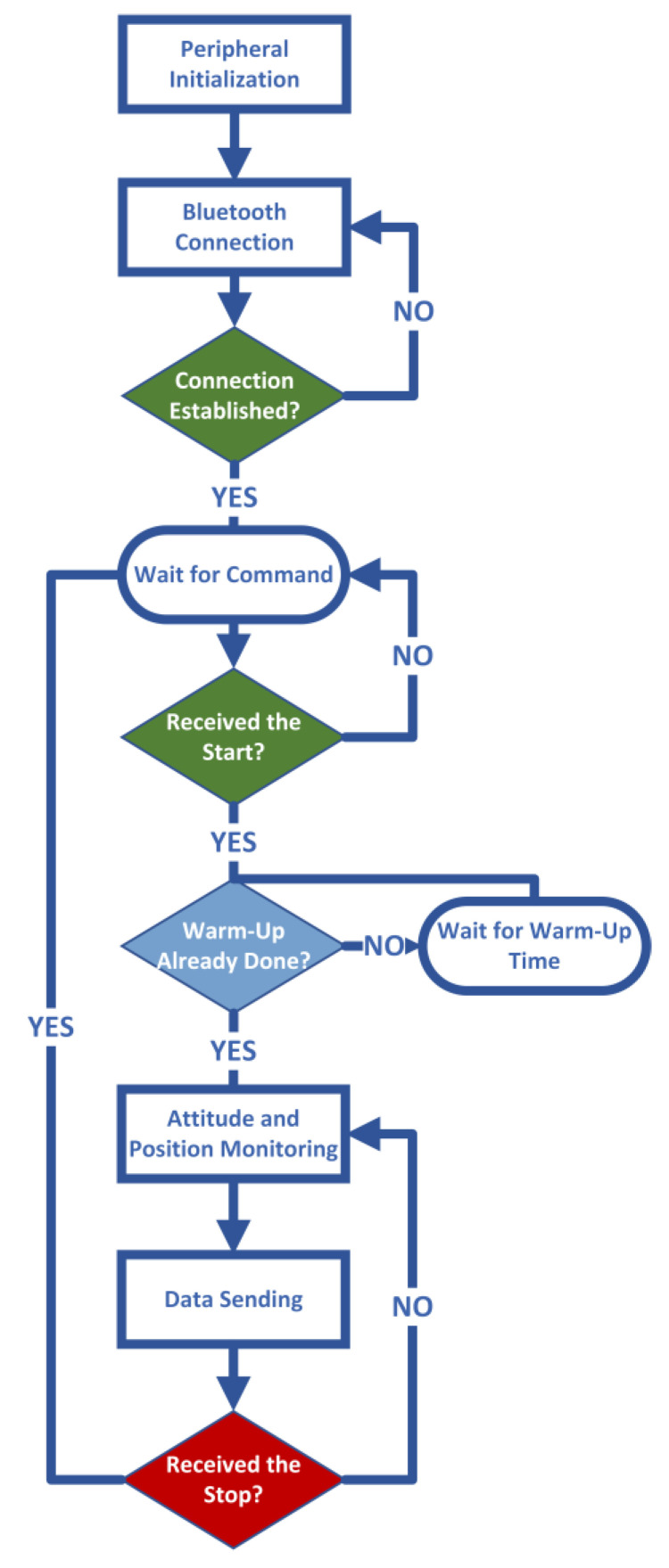
Microcontroller program flow chart.

**Figure 4 sensors-22-01788-f004:**
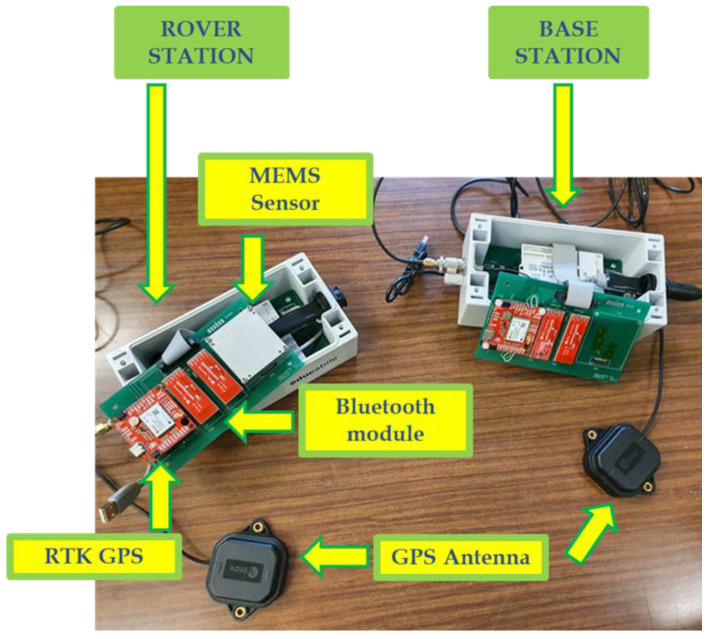
Realized prototype for monitoring the attitude and position of large structures.

**Figure 5 sensors-22-01788-f005:**
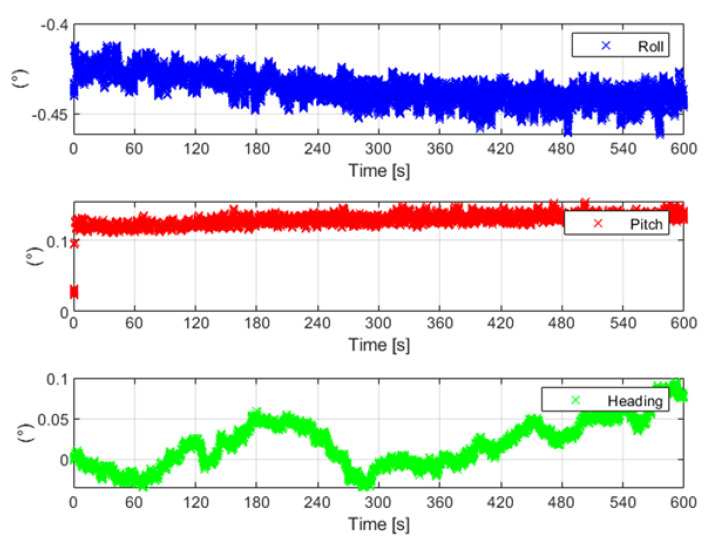
Attitude estimates under stationary conditions.

**Figure 6 sensors-22-01788-f006:**
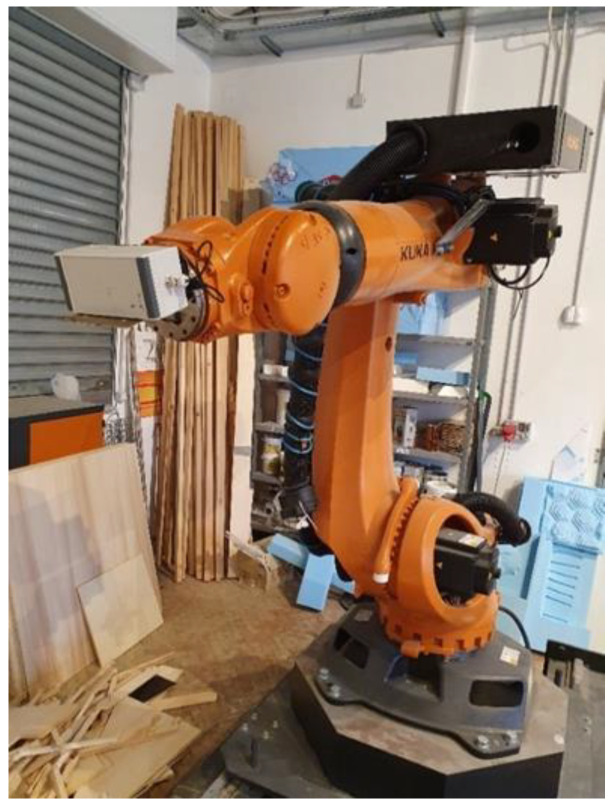
Test setup for system output stability.

**Figure 7 sensors-22-01788-f007:**
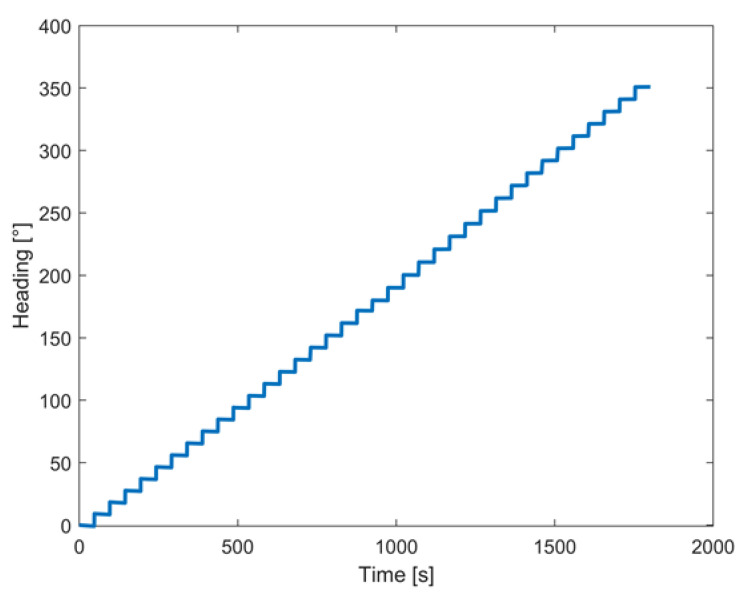
Measured angles from 0° to 360° with a step of 10°.

**Figure 8 sensors-22-01788-f008:**
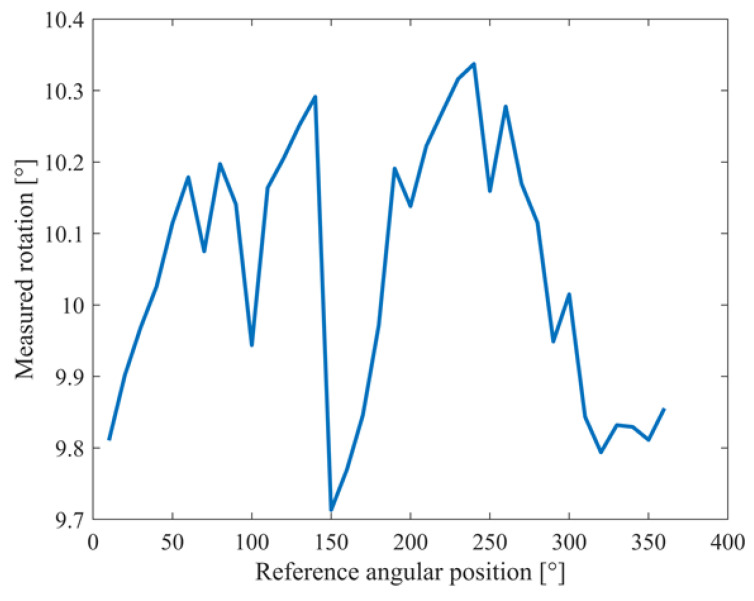
Measured rotations for different angular positions.

**Figure 9 sensors-22-01788-f009:**
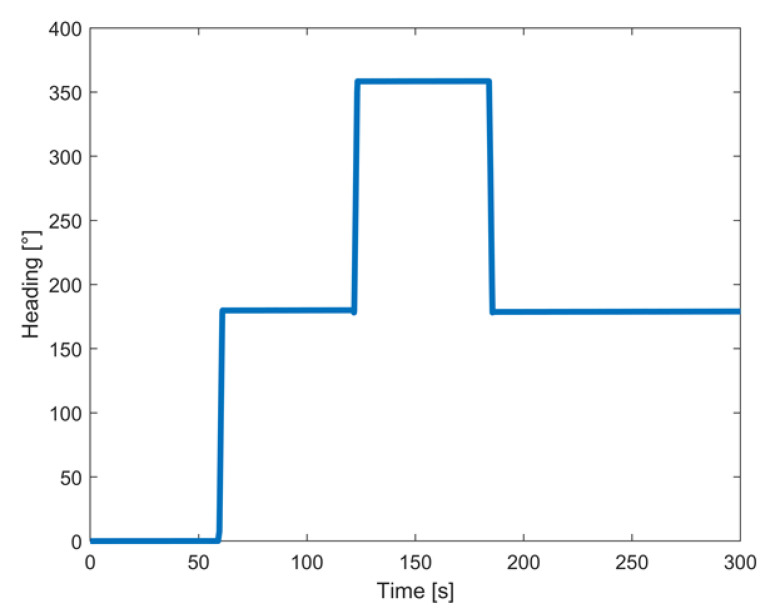
Heading values in controlled rotation from 0 to 360 degrees in four steps.

**Figure 10 sensors-22-01788-f010:**
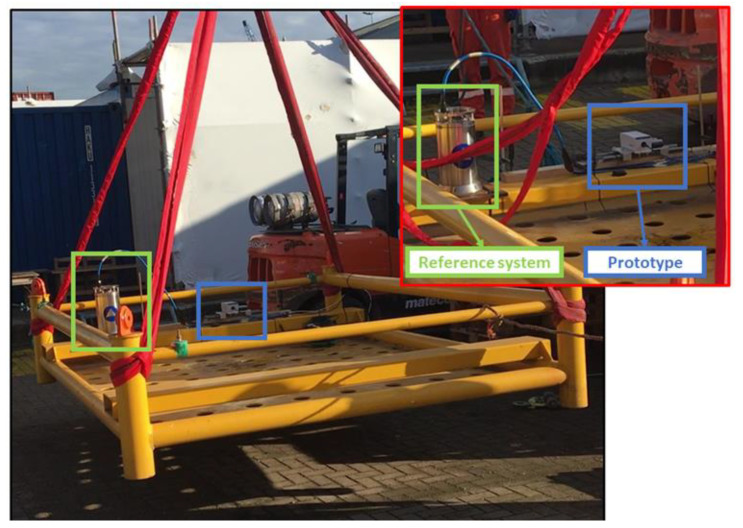
Prototype and reference system installed on the oscillating platform exploited for comparison tests.

**Figure 11 sensors-22-01788-f011:**
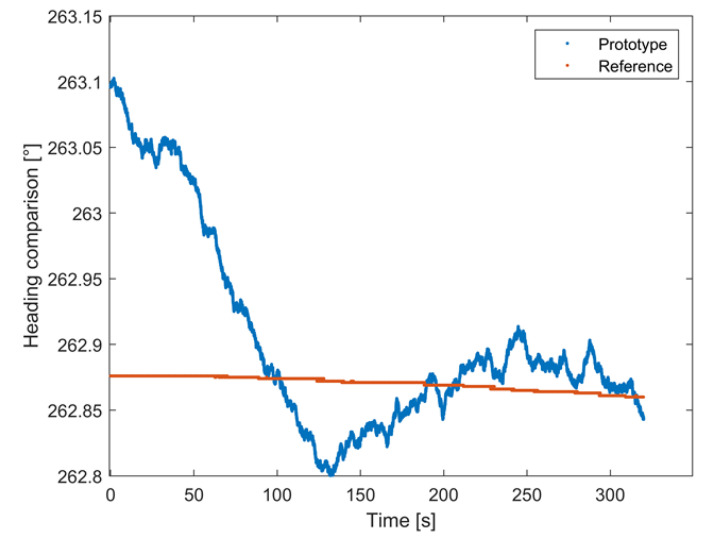
Heading angle comparison between the proposed system (blue) and FOG (red).

**Figure 12 sensors-22-01788-f012:**
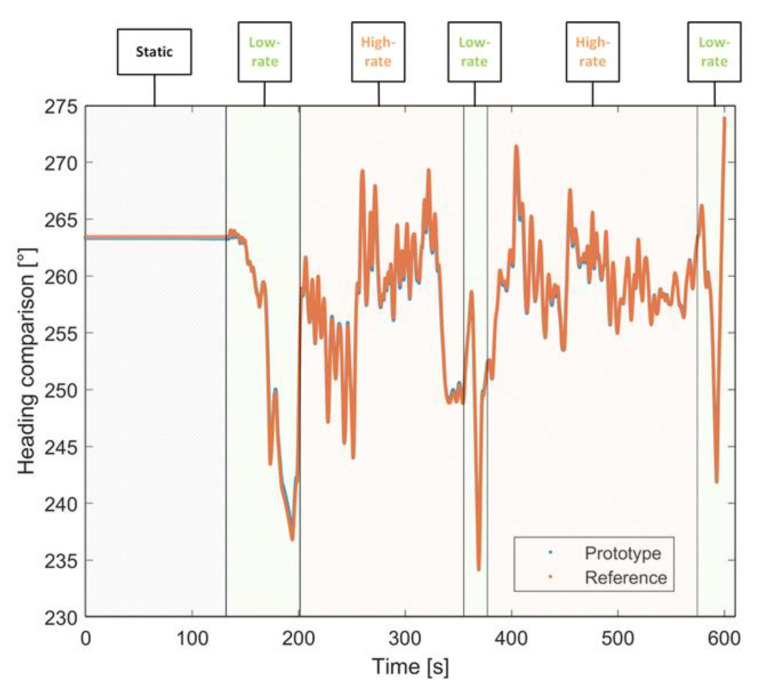
Heading angle comparison in tests conducted in dynamic conditions.

**Figure 13 sensors-22-01788-f013:**
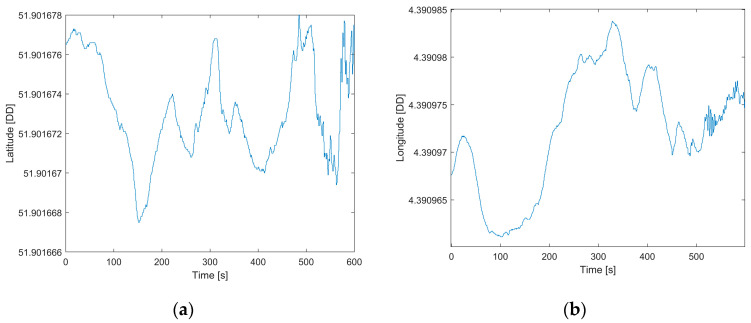
Position estimates (expressed in Decimal Degrees (DD): Latitude (**a**) and Longitude (**b**)) provided by the proposed solutions in the test conducted in dynamic conditions.

**Table 1 sensors-22-01788-t001:** Attitude estimates results.

Angle	Mean Value [°]	STD [°]	RMS [°]
Roll	0.44	0.23	0.45
Pitch	0.13	0.11	0.13
Heading	0.06	0.04	0.05

**Table 2 sensors-22-01788-t002:** Mean measured values in the presence of nominal rotations equal to 10°.

Output	Degrees (°)
Mean Value	10.04
STD	0.18

**Table 3 sensors-22-01788-t003:** Differences between the nominal and measured angles experienced in tests involving complete overturns.

Output	Degrees (°)
Mean Value	0.01
STD	0.15

**Table 4 sensors-22-01788-t004:** Results of the comparison tests in standing conditions expressed in terms of differences between the measured and reference angles.

Attitude	Mean Value (°)	STD (°)	RMSE (°)
Heading	0.03	0.07	0.08
Pitch	0.07	0.34	0.35
Roll	0.22	0.31	0.38

**Table 5 sensors-22-01788-t005:** Difference between the measured and reference heading angle expressed in relative percentage terms after Kalman filter convergence.

Angle	Mean Value (%)	STD (%)	RMS (%)	Min Value (%)	Max Value (%)
Heading	0.012	0.011	0.031	0.018	0.027

**Table 6 sensors-22-01788-t006:** Results of the comparison tests under standing conditions expressed in terms of the differences between the measured and reference angles.

Attitude	Mean Value (°)	STD (°)	RMS (°)
Heading	0.05	0.38	0.38
Pitch	0.44	0.23	0.5
Roll	0.24	0.23	0.34

**Table 7 sensors-22-01788-t007:** Difference between the measured and reference heading angle expressed in relative percentage terms in dynamic tests.

Attitude	Mean Value (%)	STD (%)	RMS (%)	Min Value (%)	Max Value (%)
Heading	0.02	0.15	0.15	0.49	0.52

**Table 8 sensors-22-01788-t008:** Performance comparisons in position estimation expressed in Decimal Degrees (DD).

Position	Reference (DD)	Mean Value (DD)	Δ Position (DD)
Latitude	51.9016731	51.9016746	−1.4400 × 10^−6^
Longitude	4.390979020	4.390974119	4.9010 × 10^−6^

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
