# Peer review of "Low-Cost and High-Performance Solution for Positioning and Monitoring of Large Structures"

_sensors, 2022, doi:10.3390/s22051788_

Round 1

Reviewer 1 Report

Much technical literature deals with GPS positioning using a Base station. In this article, this method was used in conjunction with the use of a gyroscopic sensor to accurately determine the position of a large device. This is an interesting technical solution.

To the detriment of the matter, the fact that they dealt with the determination of the instantaneous position of, for example, vibrating objects (tower, bridge, mast, ...).It is desirable that the article contains a flowchart of the microcontroller program and a description of the function of the device. 

Author Response

Dear Reviewer, Editors and Guest Editors,

we want to thank you for the detailed suggestions and comments regarding the paper titled `Low-Cost and High-Performance Solution for Positioning and Monitoring of Large Structures'. We have addressed all the reported concerns and updated the paper accordingly. Modifications and updates in the paper have been highlighted by means of “Track change” function and are summarized in the attached file.

Paper quality has been greatly improved and we hope that you now find it suitable for publication. If you have any further queries, please do not hesitate to get in contact with us.

Yours sincerely,

Giorgio de Alteriis

Reviewer 2 Report

This paper illustrated a  low-cost  positioning  system based on the  Kalman filter algorithm for MEMS sensors. The overall paper is well organized, but some important points should be considered before publish.

  1. The descriptions of main algorithms were too simple in current form. For example, how did the authors combine two modifications, namely "ba,bg" and "dV,dP,dbias"  in one mechanization equations in figure 1. What are the (or Are there any) differences of the mechanization equations in this paper from that in the ref 15? Also, I suggest the authors should give more details about the Kalman filter applied in this paper. Without these detail descriptions, other readers cannot reproduce the methods and the results of this paper.
  2.  I don't think figure 11 is necessary. In my opinion, it just the subtraction of blue line and red line in figure 10. A table shows the mean value and STD value of the differences is enough. The same as figure 13 and figure 14. I suggest the authors use data and tables to compare there results to commercial FOG and emphasize their advantages.

Author Response

(The authors gave the same response as above.)

Round 2

Reviewer 2 Report

I suggest that the manuscript can be accepted according to these modifications.